# *Listeria monocytogenes* infection rewires host metabolism with regulatory input from type I interferons

Duygu Demiroz[1,2], Ekaterini Platanitis[1], Michael Bryant[1], Philipp Fischer[1], Michaela Prchal-Murphy[3], Alexander Lercher[4,5], Caroline Lassnig[6,7], Manuela Baccarini[1], Mathias Müller[6,7], Andreas Bergthaler[4], Veronika Sexl[3], Marlies Dolezal[3], Thomas Decker[1]*

1 Department of Microbiology, Immunobiology and Genetics, Max Perutz Labs, University of Vienna, Vienna Biocenter, Vienna, Austria, 2 Vienna BioCenter PhD Program, a Doctoral School of the University of Vienna and the Medical University of Vienna, Vienna, Austria, 3 Platform for Bioinformatics and Biostatistics, Department of Biomedical Sciences, University of Veterinary Medicine Vienna, Vienna, Austria, 4 CeMM Research Center for Molecular Medicine, Austrian Academy of Sciences, Vienna, Austria, 5 Laboratory of Virology and Infectious Disease, The Rockefeller University, New York City, New York, United States of America, 6 Institute of Animal Breeding and Genetics, University of Veterinary Medicine Vienna, Vienna, Austria, 7 Biomodels Austria, University of Veterinary Medicine Vienna, Vienna, Austria

* thomas.decker@univie.ac.at

**Data Availability Statement:** The authors confirm that all data underlying the findings are fully available without restriction. All RNAseq data are

## Abstract

*Listeria monocytogenes* (*L. monocytogenes*) is a food-borne bacterial pathogen. Innate immunity to *L. monocytogenes* is profoundly affected by type I interferons (IFN-I). Here we investigated host metabolism in *L. monocytogenes*-infected mice and its potential control by IFN-I. Accordingly, we used animals lacking either the IFN-I receptor (IFNAR) or IRF9, a subunit of ISGF3, the master regulator of IFN-I-induced genes. Transcriptomes and metabolite profiles showed that *L. monocytogenes* infection induces metabolic rewiring of the liver. This affects various metabolic pathways including fatty acid (FA) metabolism and oxidative phosphorylation and is partially dependent on IFN-I signaling. Livers and macrophages from *Ifnar1*$^{-/-}$ mice employ increased glutaminolysis in an IRF9-independent manner, possibly to readjust TCA metabolite levels due to reduced FA oxidation. Moreover, FA oxidation inhibition provides protection from *L. monocytogenes* infection, explaining part of the protection of *Irf9*$^{-/-}$ and *Ifnar1*$^{-/-}$ mice. Our findings define a role of IFN-I in metabolic regulation during *L. monocytogenes* infection. Metabolic differences between *Irf9*$^{-/-}$ and *Ifnar1*$^{-/-}$ mice may underlie the different susceptibility of these mice against lethal infection with *L. monocytogenes*.

## Author summary

Many immune cells undergo metabolic remodeling following encounters with cytokines or pathogenic insults. This is essential to perform their downstream effector functions and eradicate the infectious agents. Drug-mediated interference with metabolic remodeling can have a strong impact on clearance of the pathogen. Here we describe metabolic

available: https://www.ncbi.nlm.nih.gov/geo/query/acc.cgi?acc=GSE162448.

**Funding:** Funding was provided by the Austrian Science Fund (FWF) through grants P25186-B22, SFB F6101 (to MM, CL and MD) and SFB F6103 (to TD). DD was supported through a unidoc fellowship from the University of Vienna. AL was supported by a DOC fellowship by the Austrian Academy of Sciences. This project was supported by the European Research Council (ERC) under the European Union's Horizon 2020 research and innovation program (grant agreement 677006, "CMIL" awarded to AB). The funders had no role in study design, data collection and analysis, decision to publish, or preparation of the manuscript.

**Competing interests:** The authors declare no competing interests.

changes occurring during *Listeria monocytogenes* (*L. monocytogenes*) infection of murine hosts. Infected animals show profound changes of liver metabolism that include increased glycolysis, alterations in TCA cycle metabolites and the ratio of free versus conjugated fatty acids. Similar to what has been described during viral infections, type I interferons (IFN-I), a family of cytokines produced during *L. monocytogenes* infections, are involved in the import of fatty acids (FA) into the mitochondria to generate energy via oxidative phosphorylation. Accordingly, in the absence of IFN-I signals, the cells oxidize less FAs and this might help the cells better fight the infection. We also speculate that infected cells instead boost glutamine utilization to supply their energy need. Our work describes metabolic rewiring that takes place during *L. monocytogenes* infection and the contribution of IFN-I signaling. Our study improves the understanding of listeriosis and has the potential to help us discover new drug targets against *L. monocytogenes* and viruses that induce IFN-I response.

## Introduction

Food-borne bacteria cause hundreds of thousands of deaths every year [1]. Several of these pathogenic species induce metabolic changes in the host that determine the establishment of protective immunity [2]. *Listeria monocytogenes (L. monocytogenes)* is a prime representative of this group of pathogens. *L. monocytogenes* infection induces metabolic shifts including a reduction of fatty acid oxidation (FAO) and glycolysis [3] and also anorexia [4] in *Drosophila melanogaster*. Anorexia activates a PPAR-α-driven ketogenic program, and this protects murine hosts during bacterial infection [5]. Furthermore, blocking glycolysis in mice via 2-deoxyglucose (2-DG) protects against detrimental effects of *L. monocytogenes* infection [6,7]. Thus, available experimental evidence strongly suggests metabolic reprogramming as an important facet of the host response to *L. monocytogenes* infection and as one of the parameters determining its outcome.

Following recognition and signaling via pattern recognition receptors (PRR), *L. monocytogenes* causes host cells to produce and release several antimicrobial molecules, cytokines and chemokines, including type I interferons (IFN-I) [8,9]. Engagement of IFN-I with their receptor, a heterodimer of IFNAR1 and IFNAR2 chains, initiates a signaling cascade culminating in activation of the downstream transcription factor ISGF3 (IFN stimulated gene factor 3) and its association with target gene promoters. ISGF3 is assembled from signal transducers and activators of transcription 1 and 2 (STAT 1 and 2) as well as IFN regulatory factor 9 (IRF9), the DNA-binding component of the complex. ISGF3 activity contributes to a robust innate immune response against the invading pathogen. However, unlike their well-known protective effects against viruses, IFN-I have an adverse effect on innate immunity to *L. monocytogenes* infection, decreasing the resistance of mice against lethal infection [8,10].

Classical M1 activation of macrophages via Toll-like receptor signaling induces glycolysis and FA synthesis (FAS) which are required for enhanced effector functions [11]. In contrast, FAO is a hallmark of alternatively activated macrophages and crucial for their tolerogenic functions. IFN-I were shown to promote glycolysis and oxidative phosphorylation (OXPHOS) in plasmacytoid dendritic cells (pDCs) [12]. The authors of this study concluded that the OXPHOS in CpG-stimulated pDCs is mainly driven by fatty acid oxidation (FAO), which is dependent on IFNAR signaling. In high-fat-diet models IRF9 was shown to regulate expression of fatty acid (FA) metabolism genes and in this way protect from hepatic steatosis and insulin resistance [13]. Together, these studies support the conclusion that IFN-I and their

signal transducers regulate metabolism, adding yet another activity to their well described roles in antimicrobial and inflammatory responses.

Once *L. monocytogenes* crosses the intestinal barrier, it reaches the liver, the body's major metabolic hub. Reportedly, *L. monocytogenes* infection reduces liver ATP and $NAD^+$ levels and causes an associated impairment of OXPHOS [14,15]. Further impact of *L. monocytogenes* on liver metabolism or a potential metabolic control by IFN-I and their transcription factor ISGF3 has not been investigated. Here we show that *L. monocytogenes* infection induces a gene expression program in the liver that affects various metabolic pathways including FA metabolism and OXPHOS and inhibition of FAO partially protects mice from detrimental effects of *L. monocytogenes* infection. Both IRF9 and IFNAR1 deficiency alter liver and macrophage metabolism during infection, but in some cases their impacts differ. Our findings indicate that IFN-I are integral to metabolic regulation during *L. monocytogenes* infection and that some of their activities do not require ISGF3/IRF9-mediated transcription.

## Results

### *Irf9*$^{-/-}$ and *Ifnar1*$^{-/-}$ mice differ in the level of protection from *L. monocytogenes* infection

IFN-I govern their target genes through formation of the ISGF3 complex. The IRF9 subunit of this complex is essential for its association with DNA. We sought to determine whether the protection from the adverse effects of *L. monocytogenes* infection of *Ifnar1*$^{-/-}$ mice [16–18], resulted entirely from ISGF3-dependent activities of IFN-I or whether ISGF3-independent effects contribute [19]. We thus infected mice that lack either IRF9 or IFNAR1 with *L. monocytogenes* for 10 days and monitored their survival rates. Consistent with previous reports, IFNAR1 deficiency enhanced the survival of infected mice. Surprisingly, *Irf9*$^{-/-}$ mice were protected to an even higher degree. (**Fig 1A**). Under the experimental parameters of our infection experiments the increase in resistance produced by IRF9-deficiency was highly significant. However, protection of *Ifnar1*$^{-/-}$ mice did not reach a 95% confidence interval although it was a highly reproducible phenotype in line with the literature [8,10,20]. Unexpectedly, this difference in survival rates between *Irf9*$^{-/-}$ and *Ifnar1*$^{-/-}$ mice was not reflected by the bacterial load of the liver and spleen three days post-infection (**Fig 1B**). Thus, the benefit of IRF9 versus IFNAR1 deficiency does not result from increased killing of the bacteria during the early innate response.

IFN-I are known to control a subset of inflammatory cytokines and chemokines. To investigate whether IRF9 and IFNAR1 deficiencies differ in the synthesis of inflammatory mediators, we profiled cytokines and chemokines released into the blood after *L. monocytogenes* infection. Infection led to increases in proinflammatory CCL2, IFNγ, IL6 and CXCL10 in wt mice. (**Figs 1C and S1A**). Furthermore, both IRF9 and IFNAR1 deficiencies reduced the amount of IL6 and the monocyte attractant CCL2, with CCL2 being affected more in absence of IRF9 than IFNAR1. Consistently, we found reduced inflammatory monocyte recruitment to the peritoneal cavity of infected *Irf9*$^{-/-}$ mice (**Fig 1D**) while differences between neutrophil recruitment were not significant (**S1B and S1C Fig**). Monocyte recruitment is considered a protective component of the innate response to *L. monocytogenes* [21]. Thus, reduced monocyte recruitment fails to explain the increased protection observed in IRF9-deficient mice.

The death of splenic T cells and hepatocytes was shown to increase the susceptibility of mice to *L. monocytogenes* infection [22,23]. To assess the contribution of IFN-I and IRF9 to the cytotoxic effect of infection and liver damage, we measured serum levels of alanine aminotranferase (ALT). Both knockouts were similarly protected from *L. monocytogenes*-induced liver damage (**Fig 1E**). Furthermore, we quantified the total splenocyte death in infected mice.

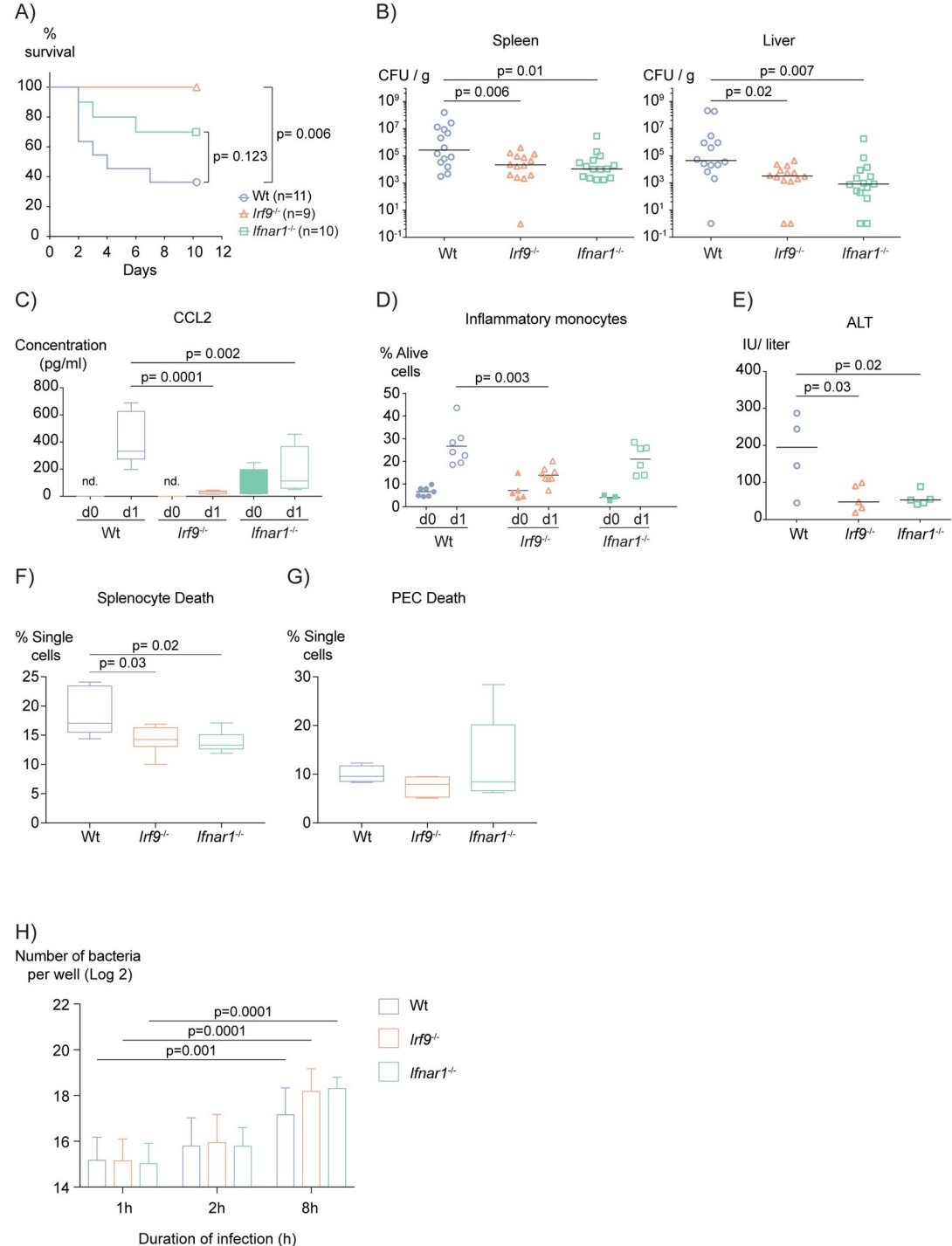

**Fig 1. _Irf9_⁻/⁻ and _Ifnar1_⁻/⁻ mice differ in the level of protection from _L. monocytogenes_ infection.** (A) Kaplan-Meier plots showing survival of mice after infection with 10⁶ _L. monocytogenes_ for 10 days. P values were calculated using log-rank test and corrected for multiple testing using Bonferroni-Holm correction. (B) Bacterial loads of spleen and liver three days post-infection. (C) CCL2 levels measured from the plasma of mice one day after PBS injection or _L. monocytogenes_ infection using Luminex-based multiplex bead array. The figure shows the pool of two experiments (Number of uninfected mice used wt = 10; _Irf9_⁻/⁻ = 10; _Ifnar1_⁻/⁻ = 8 and one-day infected mice wt = 9; _Irf9_⁻/⁻ = 9; _Ifnar1_⁻/⁻ = 8) (D) Flow cytometric analysis of inflammatory monocytes derived from peritoneal lavage of mice one day after PBS injection or _L. monocytogenes_ infection. (E) Alanine aminotransferase (ALT) levels measured in plasma three days post-infection. (F) Percentage of total splenocyte (n = 6 for all genotypes) and (G) PEC (n = 4, 5 and 6, respectively) death gated from single cells of mice one day post-infection, quantified

using flow cytometry. Each data point represents one biological replicate. Minimum number of mice used per condition is four. The median values are shown with lines. Whiskers show 5–95 percentile. P values for CFU graphs were calculated using log10 transformed CFU values. For all statistical analysis ANOVA was performed and corrected for multiple testing with Tukey or Dunnett's post-hoc test. (H) *In vitro* CFU assay showing log 2 transformed number of viable *L. monocytogenes* after 1h, 2h or 8h of infection per well. Each data point represents one biological replicate. Bars show the mean values. P values were calculated using two-way ANOVA and corrected for multiple testing with Dunnett's post-hoc test. n represents the number of mice used for the specified experiment. nd: not detected.

Similar to liver damage, animals from both knockout mice strains showed diminished splenocyte death compared to wt mice (**Fig 1F**). In contrast, the inflammatory infiltrate in the peritoneal cavity of the same mice did not show differences in cell viability between wt and gene-deficient mice (**Fig 1G**). Collectively, these results show that IFN-I signaling augments liver damage and splenocyte death in *L. monocytogenes*-infected mice with similar contributions of IFNAR1 and IRF9.

Macrophages are the most important effector cells in *L. monocytogenes* infection [8,24]. In line with the literature [17], we detected no change in the number of intracellular viable *L. monocytogenes* in both *Irf9*[-/-] and *Ifnar1*[-/-] primary bone-marrow-derived macrophages (BMDMs) compared to wt. This suggests similar killing potential for *L. monocytogenes* (**Fig 1H**). Collectively, these data show that the implications accounting for the detrimental effects of IFN-I, such as IFN-I-induced liver damage and splenic cell death, show no difference between the knockouts. Furthermore, defective monocyte recruitment cannot account for the protection. Therefore, these parameters cannot explain the ISGF3-independent role of IFN-I in the fight against *L. monocytogenes*.

## IFN-I signaling interferes with glutaminolysis and enhances FAO in macrophages

Macrophages are critical in establishing the first line of defense against *L. monocytogenes*. Influence of IFN signaling on the metabolism of immune cells is well established [25]. Mining of recent macrophage RNAseq data [26] showed that metabolic pathways such as fatty acid (FA) metabolism and oxidative phosphorylation (OXPHOS) gene sets are among the most differentially expressed gene sets in *Irf9*[-/-] macrophages treated with IFN-I (**S2A Fig**). To detect whether wt, *Irf9*[-/-] and *Ifnar1*[-/-] BMDMs experienced changes in FA metabolism when infected with *L. monocytogenes*, we performed targeted liquid chromatography-tandem mass spectrometry (LC-MS) measurements of free FA. Intracellular FA levels (**S2B Fig**), as well as expression of *Cd36*, a protein responsible for FA uptake (**S2C Fig**), were not affected by the loss of IRF9 or IFNAR1. Acetyl-CoA Carboxylase (ACC) is responsible for conversion of acetyl CoA into malonyl CoA and its activity is inhibited by phosphorylation. Malonyl CoA is the first substrate of *de novo* lipogenesis and it inhibits CPT1, the enzyme responsible for conjugation of FAs to carnitine, thereby, inhibiting FA oxidation (FAO) [27]. Intensities of phospho-ACC (pACC) bands obtained by western blotting were decreased in both *Irf9*[-/-] and *Ifnar1*[-/-], with infected *Ifnar1*[-/-] BMDMs bearing significantly less pACC (**Figs 2A and S2D**). FAs are imported into the mitochondria via carnitine for FAO. Whereas conjugated carnitine (stearoyl carnitine) is engaged in FA import into the mitochondria, free carnitine is not. Our targeted LC-MS measurement for these metabolites showed that *Ifnar1*[-/-] BMDMs had lower stearoyl carnitine after infection, despite similar levels of free carnitine compared to wt (**Fig 2B**). This suggests that less FAs are transported into the mitochondria for oxidation in the absence of IFN-I signaling.

Seahorse flux analysis is a tool to assess cellular OXPHOS via oxygen consumption rate (OCR). OXPHOS is fueled mainly by oxidation of glucose, amino acids and FA. In addition,

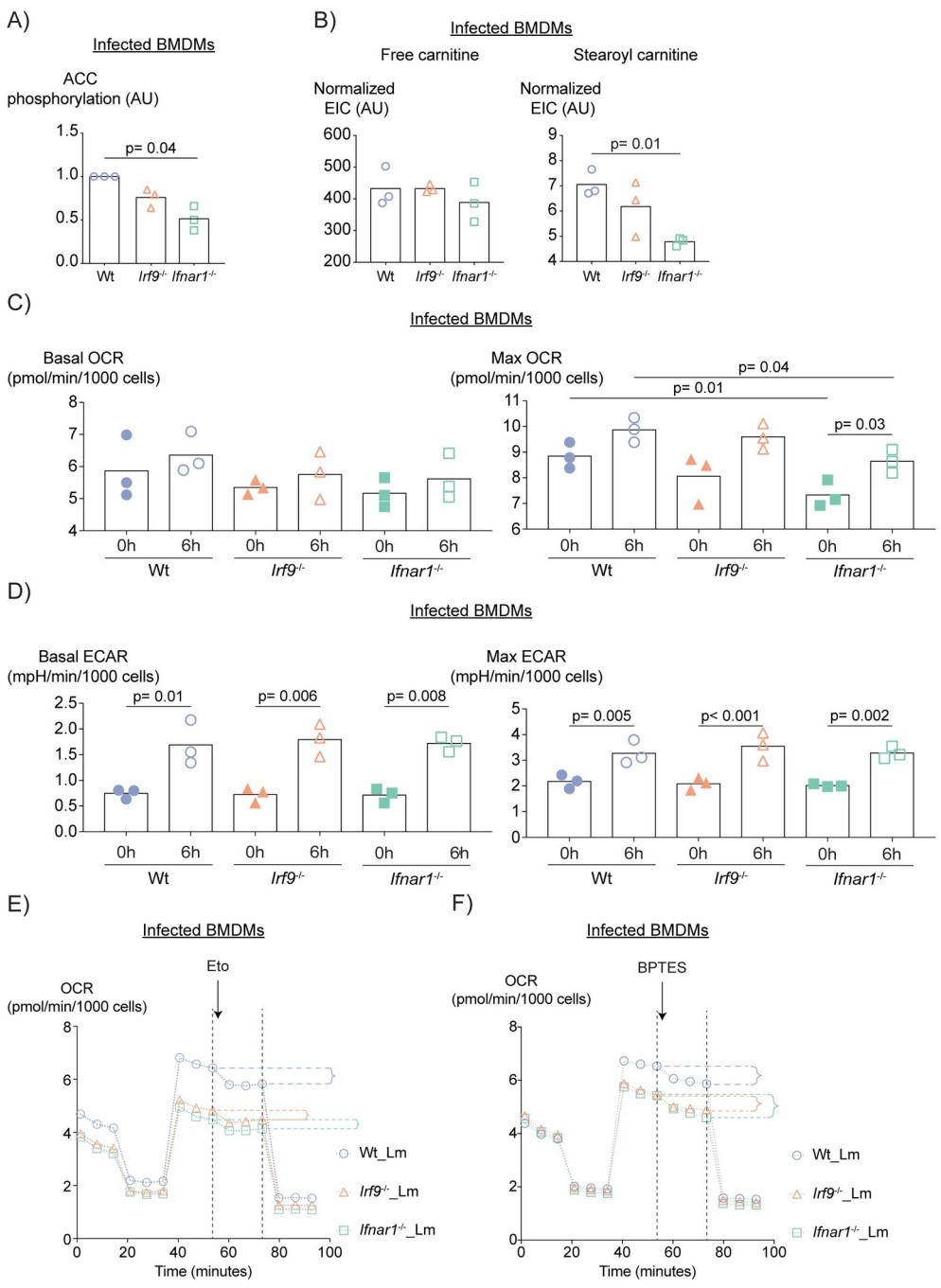

**Fig 2. *L. monocytogenes* infection and Ifnar1 deficiency lead to glutaminolysis and FAO changes in BMDMs.** (A) Quantification of ACC phosphorylation of BMDMs 6h post-infection. P value was calculated using ANOVA with Geisser-Greenhouse correction. (B) Extracted ion count (EIC) levels of carnitine and stearoyl carnitine normalized to tubulin levels as detected by WB of BMDM lysates 6h post-infection. P value was calculated using ANOVA corrected for multiple testing with Tukey's post-hoc test. (C) OCR (Oxygen Consumption Rate) and (D) ECAR (Extracellular Acidification Rate) of wt, *Irf9*[-/-] and *Ifnar1*[-/-] BMDMs uninfected or 6h post- infection. Each data point represents one biological replicate. Bars show the mean values. P values were calculated using two-way ANOVA and corrected for multiple testing with Sidak and Dunnett's post-hoc tests. Bars show the mean values. Mean OCR of four replicates of (E) Etomoxir- and (F) BPTES-treated *L. monocytogenes* infected BMDMs. The inhibitors were added at the time point indicated by the arrow. OCR differences (shown with braces) were calculated using OCR values measured at 73 min (addition of Rotenone/Antimycin) and 53 min (indicated with dashed lines).

glycolytic rates are determined by measurement of extracellular acidification rates (ECAR). In line with less FA import into the mitochondria in *Ifnar1*$^{-/-}$, Seahorse flux analysis demonstrated that infection induced an enhancement in OXPHOS in all genotypes, however, *Ifnar1*$^{-/-}$ macrophages had lower OXPHOS than wt before and after infection (**Figs 2C and S2E**). In contrast, *L. monocytogenes* infection increased glycolysis in each genotype without an impact of IRF9 or IFNAR1 (**Figs 2D and S2F**). Taken together, our data led us to assume that despite similar FA levels in all genotypes, transport of FA into the mitochondria for oxidation is enhanced by IFNAR signals. These effects were much less prominent in *Irf9*$^{-/-}$ macrophages, suggesting the contribution of an IRF9-independent pathway. To strengthen our interpretation, we measured the OXPHOS of uninfected or *L. monocytogenes*-infected wt, *Irf9*$^{-/-}$ and *Ifnar1*$^{-/-}$ BMDMs after inhibiting CPT1 with etomoxir. We treated the cells with etomoxir at the maximal OCR after uncoupling OXPHOS with FCCP (Carbonyl cyanide-4 (trifluoromethoxy) phenylhydrazone). Then, we determined the drop in OCR caused by etomoxir prior to complete shut-down of respiration with Rotenone/Antimycin. We observed less reduction in max OCR, i.e., less FAO-dependent OXPHOS, in *L. monocytogenes*-infected *Ifnar1*$^{-/-}$ but not in *Irf9*$^{-/-}$ BMDM. This is consistent with an IRF9-independent contribution of IFN-I to the induction of FAO. (**Figs 2E, S3A and S3C**). As OXPHOS was downregulated in the *Ifnar1*$^{-/-}$ BMDMs (**Fig 2C**), we also analysed the contribution of glutaminolysis, a mechanism of TCA cycle anaplerosis. To this end, we measured OXPHOS after addition of the glutaminolysis inhibitor BPTES. The drop in glutaminolysis-dependent OCR was slightly more pronounced in *Ifnar1*$^{-/-}$ compared to wt BMDMs after infection (**Figs 2F, S3B and S3D**). These data allow us to conclude that *L. monocytogenes*-infection-induced IFN-I lead to increased FAO which may be balanced by a reduction in glutaminolysis.

## Transcriptome changes in *L. monocytogenes*-infected mouse livers demonstrate regulation of genes related to FA metabolism and OXPHOS

IFN-I were recently identified as major determinants of metabolic changes occurring in hepatocytes during lymphocytic choriomeningitis virus (LCMV) infection [28,29]. This, together with the liver being a main target organ of *L. monocytogenes* infection, prompted us to investigate whether and how infection of wt, *Irf9*$^{-/-}$ or *Ifnar1*$^{-/-}$ mice changed hepatocyte transcriptomes and, particularly, genes related to metabolic pathways. We carried out RNA sequencing analyses of livers harvested from wt, *Irf9*$^{-/-}$ and *Ifnar1*$^{-/-}$ mice that were injected with PBS or *L. monocytogenes*. Gene set enrichment analysis (GSEA) comparing untreated wt with untreated *Irf9*$^{-/-}$ or *Ifnar1*$^{-/-}$ mouse liver transcriptomes resulted in the enrichment of IFN-I-stimulated genes (ISG) in the wt mouse livers, in line with a role for low-level IFN-I in maintaining homeostatic gene expression [26,30] (**S4A Fig**). Additionally, few metabolic pathways such as oxidative phosphorylation (OXPHOS), fatty acid (FA) metabolism, adipogenesis, cholesterol homeostasis and glycolysis were affected by the lack of IRF9 or IFNAR1 (**S1 and S2 Tables**). Interestingly, under homeostatic conditions only 36% (580 out of 1595) of the genes enriched in *Ifnar1*$^{-/-}$ compared to wt were common with *Irf9*$^{-/-}$ enriched genes (**S4B Fig**) pointing to a role of IFN-I signaling independent of IRF9 activity in mouse liver.

Upon infection, wt livers showed enrichment of 1390 genes (**S3 Table**). In addition to the inflammatory and interferon response pathways, these included OXPHOS, FA metabolism, glycolysis and adipogenesis hallmark genes in wt livers (**Fig 3A**), suggesting pronounced metabolic changes upon infection. Strikingly, when comparing gene sets in *Irf9*$^{-/-}$ or *Ifnar1*$^{-/-}$ with wt after infection, we identified OXPHOS and FA metabolism gene enrichment in *Irf9*$^{-/-}$ and *Ifnar1*$^{-/-}$ (**S4 and S5 Tables and Fig 3B–3D**). However, only 67.4% (93 out of 138) of enriched OXPHOS genes and 48.5% (50 out of 103) of enriched FA metabolism genes were shared

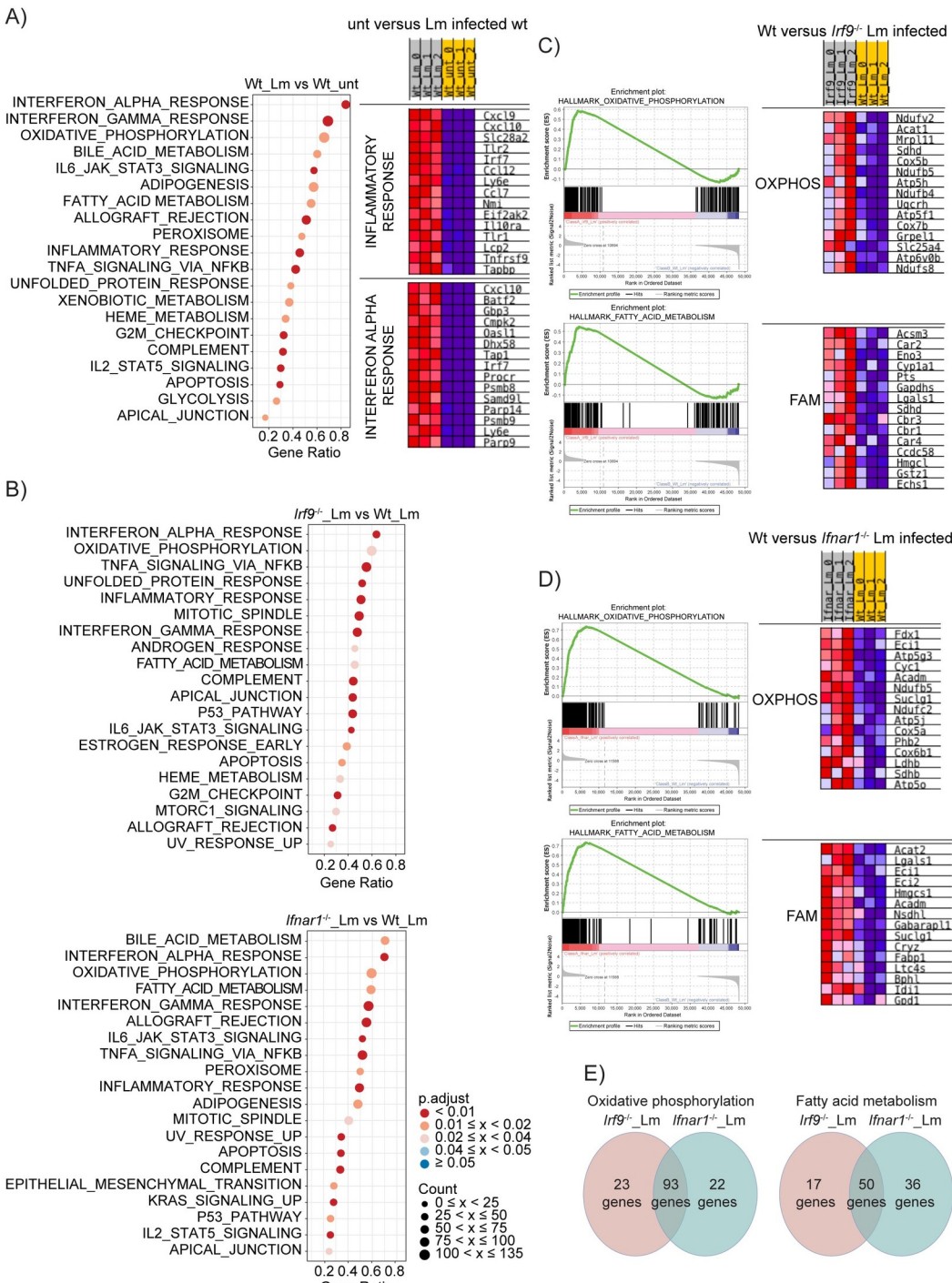

**Fig 3. Transcriptome changes in *L. monocytogenes*-infected mouse livers demonstrate regulation of genes related to FA metabolism and OXPHOS.** GSEA of RNAseq data from livers of mice one day post-infection. Dot plots of GSEA results showing the first 20 enriched pathways in (A) infected compared to uninfected wt with the heatmaps of top 15 genes with the highest enrichment score of inflammatory response and IFN alpha response pathways, (B) infected *Irf9*[-/-] and *Ifnar1*[-/-] compared to infected wt. Color and size of the dots represent adjusted p values and the number of the genes represented from a certain gene set, respectively. The x-axes were calculated by the sum of the core-enriched genes divided by its set size and ordered decreasingly. The y-axis represents the corresponding gene set name. GSEA enrichment plots for OXPHOS and FA metabolism of infected (C) *Irf9*[-/-] and (D) *Ifnar1*[-/-] compared to infected wt with the heatmaps of top 15 genes with the highest enrichment score of OXPHOS and FA metabolism pathways. (E) Venn diagrams showing the number of genes enriched in infected Irf9[-/-] and *Ifnar1*[-/-] compared to infected wt in OXPHOS and FA metabolism pathways. unt: untreated, Lm: *L. monocytogenes* infected.

between the *Irf9*$^{-/-}$ and *Ifnar1*$^{-/-}$ (**Fig 3E** and **S6 Table**). Interestingly, validation qPCRs showed increased expression of OXPHOS (*Cox5b*, *Atp5h*, *Suclg1*, *Ndufb5*) and FA metabolism (*Cpt1a*, *Sdha*, *Acadl*) genes in both *Irf9*$^{-/-}$ and *Ifnar1*$^{-/-}$ livers before and after *L. monocytogenes* infection (**Figs 3C, 3D, S5A and S5B**). On the other hand, ISGs (*Ifit3* and *Irf7*) displayed lowered expression in the absence of IRF9 and IFNAR1, as expected (**S5C Fig**). Although increased gene expression patterns seem to contradict the reduction of OXPHOS and FAO observed in BMDMs, it is likely to imply a positive feedback mechanism leading to decreased activity of these pathways. Our differential RNAseq analysis showed that IFN-I signaling and IRF9 regulate metabolic gene expression both under homeostasis and during *L. monocytogenes* infection in part by distinct mechanisms. This suggests that IFN-I and IRF9 control the immunometabolism of *L. monocytogenes* infection in the liver.

### *L. monocytogenes* infection and IFN-I signaling induce metabolic changes in the livers of mice

While the gene expression analysis showed alterations in metabolic genes, the changes are diverse and do not readily provide cues on how innate immunity to *L. monocytogenes* might be affected. Furthermore, the role of IFN-I signaling and IRF9 in this context is not clear. Therefore, we performed a targeted LC-MS-based metabolic screen of several metabolites including glycolytic, TCA cycle, FA metabolism, glutaminolysis intermediates and amino acids in the livers of mice. We applied principal component analysis (PCA) to the data. PC1 and PC2 showed separation of infected and uninfected samples independent of the genotype and accounted for 21% and 13% of the total variance, respectively (**Fig 4A**). Directional vectors on the loading plots (**S6A and S6B Fig**) implicate the significance of the metabolites in the clustering observed in the PCA. The bar graphs quantify the strength of contribution of each metabolite to the PC separation (**Fig 4E and 4F**). Relative changes in the amounts of the FAs (palmitic acid, oleic acid, palmitoleic acid), glucose and glycolytic intermediates (glucose-6-phosphate, fructose-6-phosphate and glyceraldehyde-3-phosphate) and amino acids (leucine, isoleucine, serine and valine) were the main drivers of this separation. This is in line with literature showing infection-induced oxidation of major energy sources such as glucose, fatty acids and amino acids [4]. We also detected clustering of three-day-infected mice in PC3 and PC4 (**Fig 4B**) showing that infection impacts on all basal carbon sources. Although PC1 and PC2 did not show a contribution to the separation of the genotypes, PC3 and PC4 separated the same dataset according to genotype while delineating 11% and 10% of the total variance, respectively (**Fig 4C and 4D**). We identified the *Ifnar1*$^{-/-}$ clustered distinctly from wt and *Irf9*$^{-/-}$, mainly because of the TCA intermediates succinate, malate and fumarate and also γ-aminobutyric acid (GABA) (**Fig 4D**), suggesting the regulation of TCA cycle intermediates and GABA levels by IFN-I independently of IRF9. This is in line with the finding of TCA cycle replenishment via the GABA shunt [31]. These data show that both *L. monocytogenes* infection and IFN-I signaling cause metabolic changes in the liver.

Although the PCA plots show the contribution of metabolites to the separation of a certain cluster, they do not depict the specific alterations of those metabolites. As the RNAseq data showed enrichment of FA metabolism genes in both *Irf9*$^{-/-}$ and *Ifnar1*$^{-/-}$ livers during infection, we compared FA levels in more detail by hypothesis testing. In line with the PCA analysis, all genotypes showed reduction of palmitic acid, palmitoleic acid and oleic acid after infection whereas stearic acid increased only in wt (**S6C Fig**). Additionally, we detected an increase in free carnitine levels upon infection that was less pronounced in the knockouts, particularly in *Irf9*$^{-/-}$ (**Fig 5A**). In contrast, there was significantly less increase of stearoyl carnitine in both knockout livers compared to wt upon *L. monocytogenes* infection. Stearoyl carnitine was

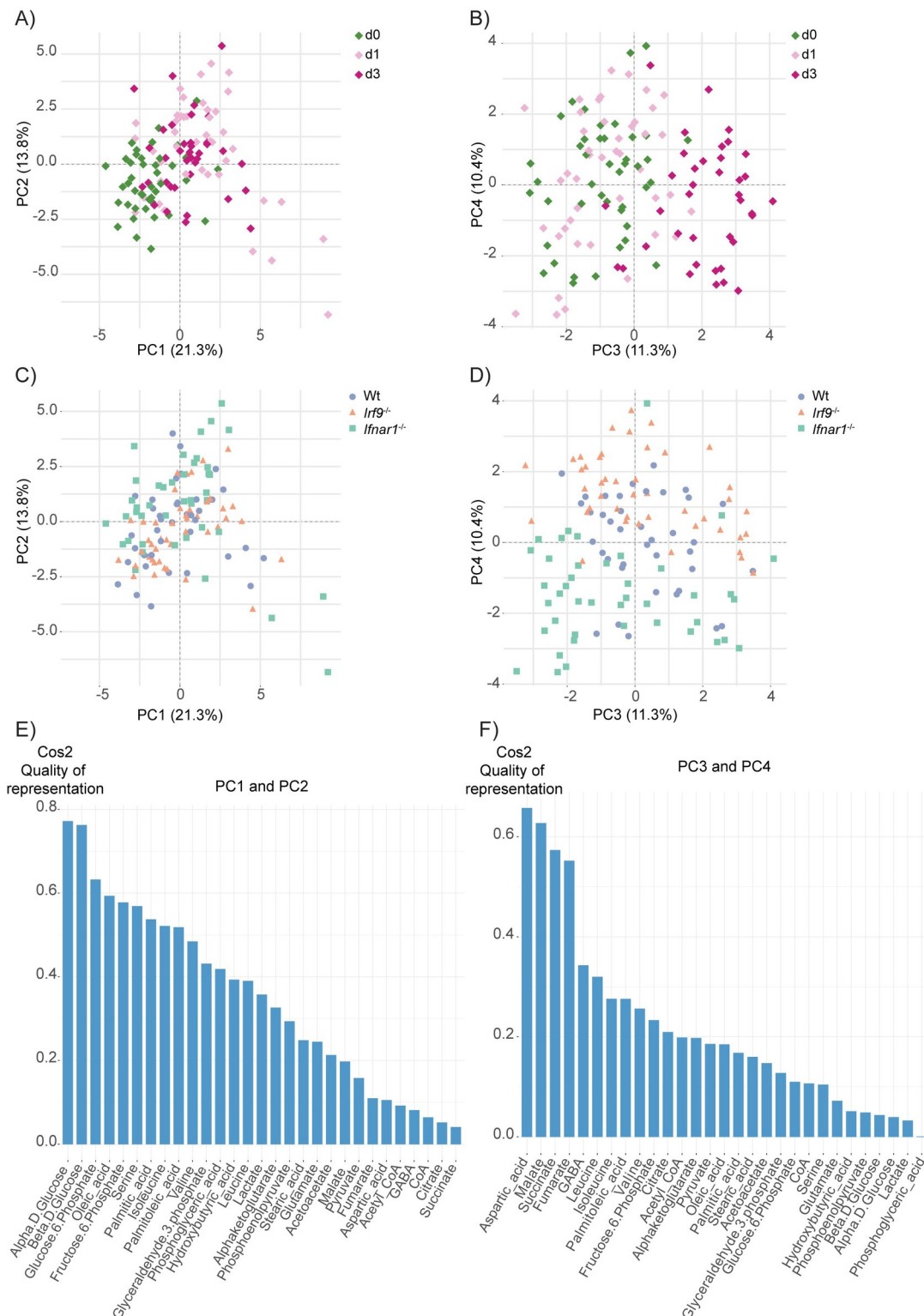

**Fig 4. _L. monocytogenes_ infection and IFN-I signaling induce metabolic changes in the liver.** PCA plots of metabolites quantified by targeted LC-MS/MS from livers of (A-B) uninfected, one-day- or three-day-infected and (C-D) wt, _Irf9_[-/-] and _Ifnar1_[-/-] mice. Percentage of total variance is indicated in the axis label. Bar graphs showing contribution of each metabolite to clustering in (E) PC1 versus PC2 and (F) PC3 versus PC4. The values are corrected for liver weight, genotype, sex of the mouse and time point effects. ANCOVA test was performed. Contrib.: contribution, PC: Principal component.

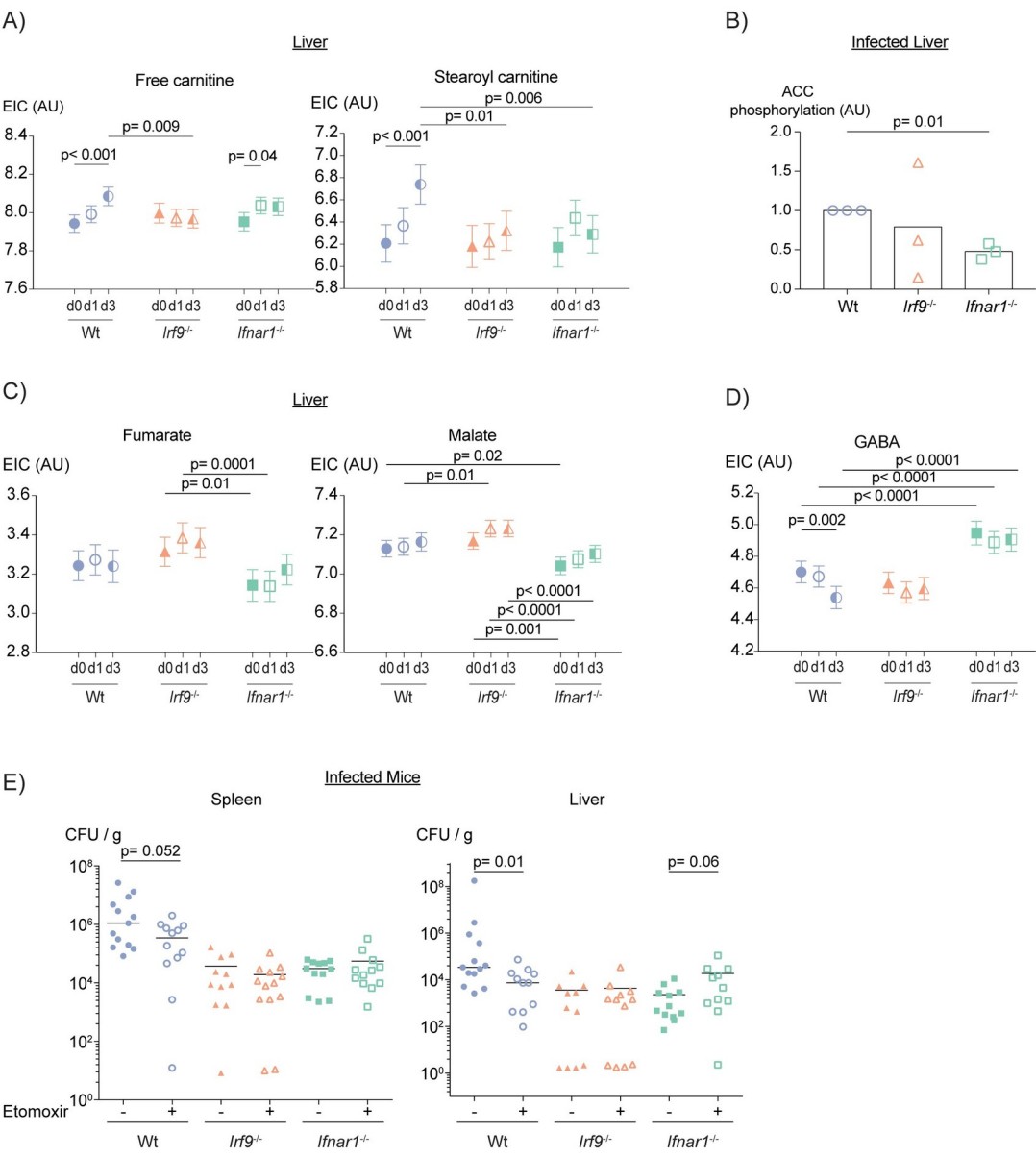

**Fig 5. *L. monocytogenes* infection and IFN-I signaling cause changes in TCA cycle, FA and glutamine metabolism intermediates of liver.** Corrected EIC (extracted ion counts) of metabolites of (A) FAO, (C) TCA cycle and (D) GABA plotted in Fig 4. The values and false discovery rate p values were calculated using values corrected for liver weight, sex of the mouse and experimental variation effects. ANCOVA test was performed. (B) Quantification of ACC phosphorylation as determined by western blot. Bars show the mean values. Each data point represents one western blot replicate of one-day infected mouse liver lysates. Error bars show minimum and maximum values. P value was calculated using ANOVA with Dunnett's and Geisser-Greenhouse correction. (E) Bacterial loads of spleen and liver of etomoxir-treated mice three days post-infection. P values were calculated using two-tailed Mann-Whitney t-test.

reduced in both *Irf9*[-/-] and *Ifnar1*[-/-] livers in three-day-infected mice providing less FA import into the mitochondria for FAO. In line with the data obtained in macrophages, western blot analysis showed less pACC in *Ifnar1*[-/-] liver lysates with *Irf9*[-/-] having only slightly less pACC **(Figs 5B and S6G)**. The data suggest that similar to *Ifnar1*[-/-] macrophages, *Ifnar1*[-/-] livers perform less FAO than wt upon *L. monocytogenes* infection, possibly due to less FA import into the mitochondria for oxidation.

Reportedly, ketogenesis induced during bacterial infection is a means of protecting the host [5] and ketone bodies (KB) are produced from FAO-derived acetyl-CoA in the liver. Based on our finding that FA metabolism is controlled by IFN-I signaling, we speculated that both knockouts might cause increased KB formation to protect the host from *L. monocytogenes* infection (**S6D Fig**). However, in both *Irf9*[-/-] and *Ifnar1*[-/-] livers the KB acetoacetate and hydroxybutyric acid were decreased three days post-infection.

Other than FA metabolism, the RNAseq data showed enrichment of OXPHOS genes in infected *Irf9*[-/-] and *Ifnar1*[-/-] mouse livers compared to wt. Furthermore, we also detected lower OXPHOS in *Ifnar1*[-/-] BMDMs. Since changes in OXPHOS are expected to alter or result from the flow of metabolites through the TCA cycle, we measured TCA cycle intermediates. While infected *Irf9*[-/-] livers contained increased fumarate and malate, *Ifnar1*[-/-] had lower levels of these metabolites (**Fig 5C**), in line with the PCA plots (**Figs 4D, 4F and S6B**). These data suggest that both IRF9 and IFN-I take part in the regulation of liver TCA metabolite levels but differentially target its metabolites.

Glutaminolysis can be used to replenish the TCA cycle by converting glutamine into α-KG or via GABA shunt (**S6E Fig**). PCA analysis suggested GABA as a reason for the separation of *Ifnar1*[-/-] livers (**Figs 4D, 4F and S6B**). Therefore, we measured the levels of intracellular GABA, the intermediate product of glutaminolysis, and found that *Ifnar1*[-/-] but not *Irf9*[-/-] had higher levels of GABA before and after infection (**Fig 5D**). We also noted a similar increase in GABA levels in uninfected *Ifnar1*[-/-] mice. Increases in expression of *Slc25a22*, the mitochondrial glutamate carrier, in *Ifnar1*[-/-] is in line with this observation (**S6F Fig**). In conclusion, our data suggest that IFN-I signaling has a role in regulating TCA cycle and glutamine metabolism in the liver. They also strengthen the conclusion that *Irf9*[-/-] and *Ifnar1*[-/-] mice differ in their impact on liver metabolism, both dependent and independent of infection with *L. monocytogenes*.

Having identified reduced FAO in BMDMs and livers of *Ifnar1*[-/-] mice and also slightly of *Irf9*[-/-] mice, we determined whether reduction of FAO provides protection against *L. monocytogenes* infection. Mice were treated with the FAO inhibitor etomoxir one day prior to and one day post-infection. [12] Our data showed that inhibition of FAO reduced the bacterial loads of spleens and livers of infected wt mice while having no or a mild effect on the knockouts (**Fig 5E**). These data lead us to conclude that FAO induced during *L. monocytogenes* infection has an adverse impact on the host and that decreased FAO in the absence of IFN-I signaling might account for part of the protection resulting from IRF9 or IFNAR1 deficiency.

## Discussion

Metabolic reprogramming has been widely recognized as an important attribute of cells engaged in various aspects of antimicrobial immunity. Many studies demonstrate that reprogramming occurs upon engagement of PRR by macrophages and by the products of PRR signaling such as IL-1 [31]. IFN-I also contribute to metabolic reprogramming. For example, a recent study of LCMV infection demonstrates that IFN-I impact on antiviral immunity by disrupting the urea cycle in the liver [29]. Here we sought to investigate whether macrophage and liver metabolism is altered by *L. monocytogenes* infection and whether any metabolic effects of *L. monocytogenes* infection are influenced by IFN-I and their transcription factor ISGF3 (**Fig 6A**).

Surprisingly, protection of *Ifnar1*[-/-] mice from *L. monocytogenes* infection is less pronounced than that provided by IRF9 deficiency. CFU assays in the target organs show that this survival variation derives from differences in resilience of mice rather than antimicrobial effector mechanisms. There are several alternative or complementary explanations for this finding.

A)

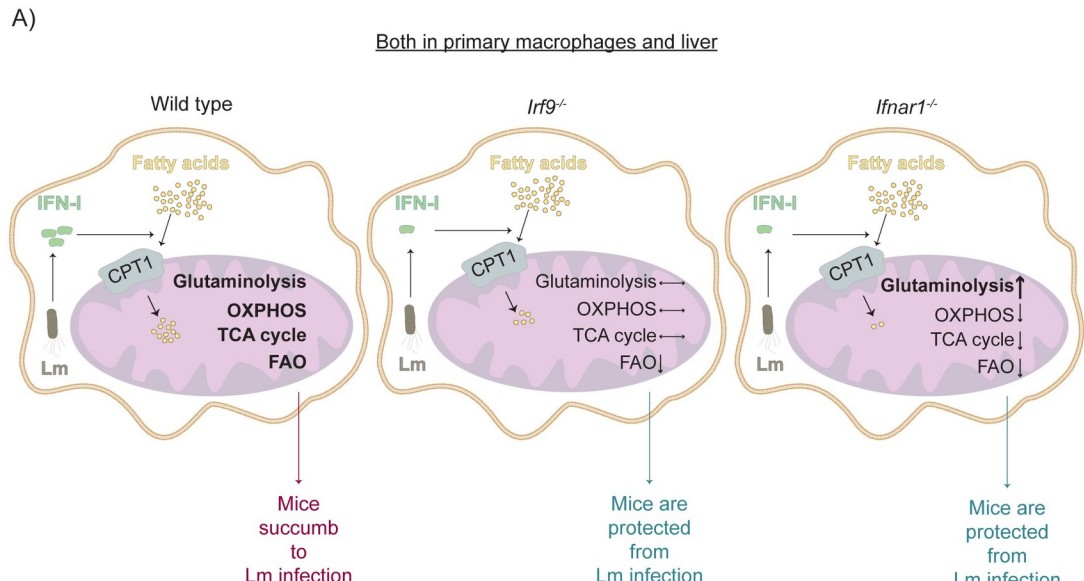

**Fig 6. Model of *Listeria monocytogenes*-induced metabolic rewiring.** (A) *Listeria monocytogenes* (Lm) induced-IFN-I regulate FA import into the mitochondria. Absence of autocrine IFNAR1 signalling leads to decreased fatty acid oxidation (FAO) in *Irf9*[-/-] and *Ifnar1*[-/-] BMDMs and livers, and also reduced TCA cycle intermediate levels and oxidative phosphorylation (OXPHOS) as well as increased glutaminolysis exclusively in *Ifnar1*[-/-] BMDMs and livers.

The most straightforward assumption is that not all immunoregulation by IFN-I signaling requires *de novo* gene transcription via ISGF3. The interpretation of our data is mainly guided by this hypothesis. The IFN-I receptor has the potential to activate MAPK and PI3K pathways [32] which may contribute to the influence of IFN-I on innate immunity to *L. monocytogenes*. For further consideration, IRF9 and IFNAR deficiency may differ in their effects on innate resistance to *L. monocytogenes* due to diverse effects on transcriptional homeostasis. This notion derives from our recent observation that IRF9 contributes to basal expression of its target genes independently of a tonic signal from the IFN-I receptor [26]. Finally, ISGF3 activity downstream of the IFN-III receptor or yet other, unexplored non-IFN-I pathways, may contribute to diverse effects of the IRF9 and IFNAR knockouts.

Mining of previous macrophage RNAseq data from our lab [26] supported a role of IRF9 in regulating FA metabolism and OXPHOS in response to IFN-I treatment. Targeted metabolite and metabolic pathway measurements in macrophages indicated *L. monocytogenes* infection to increase both glycolysis and OXPHOS. *Ifnar1*[-/-] macrophages showed a decrease in OXPHOS compared to wt. In accordance with the smallest amount of conjugated FA and the largest inhibition of ACC, we noted smallest FAO and also largest glutaminolysis dependency in *Ifnar1*[-/-] macrophages. Studies by others in IFN-I or poly I:C treated primary mouse macrophages have shown that IFN-I signaling inhibits cholesterol and FAS while enhancing lipid import for oxidation [33,34]. These changes strengthen the anti-viral response. Another study demonstrated the ability of IFN-I to induce FAO in pDCs [12]. These results concur with our data in showing a role for IFN-I in metabolic adaptation of myeloid cells to infection [25,35]. That being said, the small metabolic effects do not alone contribute to the substantial differences between *Irf9*[-/-] and *Ifnar1*[-/-] mice in resistance to *L. monocytogenes* infection, however, our research further elucidates the influence of IFN-I signaling in metabolic reprogramming.

Investigation of livers from infected mice by RNAseq and targeted metabolite measurements indicated transcriptional and metabolic reprogramming, part of which is dependent on

IFN-I signaling. Measurement of conjugated-carnitine and ACC phosphorylation showed that particularly IFNAR1 deficiency reduced FAO without affecting total intracellular FA levels. A likely explanation for this finding is that IFN-I induce FA import into the mitochondria, similar to our observations in macrophages. Reduced FAO in *Ifnar1*[-/-] livers might also explain the decreased amounts of TCA metabolites. It also in part may explain the protection from the adverse effects of *L. monocytogenes* infection, as FAO inhibition with etomoxir reduced bacterial loads in spleens and livers of wt mice albeit not to the degree observed in *Irf9*[-/-] or *Ifnar1*[-/-] mice. 80% of the liver volume is constituted by hepatocytes. Therefore, it appears safe to assume that liver RNAseq as well as the metabolite measurements reflect predominantly hepatocyte transcriptomes and metabolomes. IFN-I-dependent FAO can thus be considered a common facet of macrophage and hepatocyte metabolism and a joint contributor to the detrimental effects of IFN-I on *L. monocytogenes* infection.

Although the metabolite differences between the livers of wt, *Irf9*[-/-] and *Ifnar1*[-/-] mice are relatively low, they are in line with the gene expression patterns obtained from the same mice. Moreover, PCA plots show that infection induces clear changes in the liver metabolism of main carbon sources, namely glucose, FA and amino acids. This is again consistent with the genotype-independent glycolytic switch in macrophages. Additionally, both liver and macrophages reflect an activating effect of IFN-I on FA metabolism during *L. monocytogenes* infection. However, we speculate that the FAO reduction in both *Irf9*[-/-] and *Ifnar1*[-/-] can be compensated by glutaminolysis only in *Ifnar1*[-/-]. This hypothesis is supported by increased GABA in *Ifnar1*[-/-] livers and by the slightly increased OXPHOS dependency of *Ifnar1*[-/-] macrophages on glutaminolysis compared to wt. Of note, recent reports attribute a protective effect to GABA in both acute liver injury [36] and *L. monocytogenes* infection [37], although involvement of IFN-I signaling has not been addressed in the latter study. Therefore, we speculate that augmented GABA production in *Ifnar1*[-/-] livers might contribute to the protection of *Ifnar1*[-/-] mice from *L. monocytogenes* infection. The mechanism behind the protective effects of systemic GABA is subject to further study and metabolic tracing experiments are needed to validate the relationship between GABA synthesis and the protective effect of the *Ifnar1*[-/-].

Differences in metabolite levels and pathway outputs between *Irf9*[-/-] and *Ifnar1*[-/-] macrophages and livers are consistent with the assumption that the two genotypes produce differences in the FA metabolism. Most likely, IFN-I regulate FAO to some extent without requiring ISGF3-dependent *de novo* gene transcription. This is reflected in RNAseq data from infected livers as well. Other than the expected cytokine and inflammatory signaling pathways, we found OXPHOS and FA metabolism pathways among the most differentially regulated pathways in both *Irf9*[-/-] and *Ifnar1*[-/-] compared to wt, suggesting a metabolic dysregulation in these mice. However, *Irf9*[-/-] and *Ifnar1*[-/-] enriched gene subsets vary considerably, suggesting an IRF9-independent role for IFN-I in regulating metabolic gene expression in the liver following *L. monocytogenes* infection. To our surprise, a closer look at the expression pattern of single genes showed increased expression of the OXPHOS and FA metabolism genes in *Irf9*[-/-] and *Ifnar1*[-/-] livers both before and after infection. This led us to speculate that disturbance of these pathways results in reduced OXPHOS and FAO. This may cause a boost in gene expression due to a positive feedback response to compensate for the reduced pathway activity. Also, post-transcriptional regulatory mechanisms might contribute to the effect we have observed. However, these need to be tested in further detail. In addition, differences in FA metabolism may result from ISGF3-independent IRF9 activity. This is in line with previous studies demonstrating an IRF9 involvement in FA metabolism, insulin resistance [13], cardiac hypertrophy [38] and hepatic-ischemia-reperfusion injury [39]. These roles of IRF9 beyond IFN-I signaling may contribute to the survival differences we detected in *L. monocytogenes*-infected *Irf9*[-/-] and *Ifnar1*[-/-] mice.

In conclusion, our data establish metabolic reprogramming of both liver and macrophages upon *L. monocytogenes* infection. Although IFN-I signaling acts similarly in liver and primary macrophages in regulating FAO and glutaminolysis, the roles acquired by the hepatocytes during infection has to be further elucidated. While IRF9 is downstream counterpart and responsible for most of the of IFN-I-induced gene expression, IFN-I affect *L. monocytogenes* infection and its characteristic metabolism independently of IRF9 and vice versa in liver and macrophages. Furthermore, we show that IFN-I signaling is an essential player in the regulation of TCA metabolite levels, FAO and thus OXPHOS and inhibition of FAO has a protective role during *L. monocytogenes* infection. Understanding the metabolic pathways controlled by IRF9 and IFN-I during *L. monocytogenes* infection might lead the way to a better understanding of listeriosis.

# Materials and methods

## Ethics statement

All, wt, *Irf9*$^{-/-}$ and *Ifnar1*$^{-/-}$ mice in C57BL6/N background, were bred and maintained at the Max Perutz Labs under specific pathogen-free conditions according to Federation of European Laboratory Animal Science Associations (FELASA) guidelines. All animal experiments were approved by the ethics and animal Welfare Committee of the University of Veterinary Medicine and the national authority (Austrian Federal Ministry of Science and Research) according to §§ 26ff. of Animal Experiments Act, Tierversuchsgesetz 2012-TVG 2012 (BMWFW-68.205/0032-WF/II/3b/2014; BMWFW-68.205/0173-V/3b/2019) and conform to the guidelines of FELASA and ARRIVE (Animal Research: Reporting of In Vivo Experiments). Age and sex-matched (8–10 weeks) mice were used for all experiments.

## Cell culture

Bones harvested from mice were crashed and filtered using 70 mm cell strainer to isolate bone marrows (BM). All BM cells were grown and differentiated in DMEM + 10% FCS +100 U/ml Penicilin + 100 μg/ml Streptomycin medium with 500 ng/ml MCSF (kind gift of Löms Ziegler-Heitbrock, Helmholtz Center, Munich, Germany) for 10 days. The day before infection, macrophages were seeded in DMEM + 10% FCS + 500 ng/ml MCSF. For western blotting, qPCR and flow cytometry experiments $10^6$ cells, for seahorse flux assays $3.5 \times 10^4$ cells and for targeted metabolite measurements $10^7$ cells were seeded in one well of six-well plates, 96-well seahorse plates and 15 cm plates, respectively.

## BMDM infections

For BMDM infections the LO28 strain of *L. monocytogenes* was inoculated in brain heart infusion (BHI) medium for overnight growth. The calculated volume ($1OD_{600nm} = 5 \times 10^8$ viable bacteria) was directly pipetted into the wells with BMDMs. Medium containing *L. monocytogenes* was changed into 50 μg/ml gentamicin + DMEM + 10% FCS after 1 hour of infection, then, into 10 μg/ml gentamicin + DMEM + 10% FCS after 2 hours of infection, which was kept on the cells until the end of the experiment. For *in vitro* colony forming unit (CFU) assays, 50000 BMDMs per well were seeded in 96 well plate. The cells were washed twice with PBS after the duration of infection was over and lysed in 50 μl ddH$_2$O twice for 5 min at 37°C. Three 1:10 serial dilutions of these lysates were plated on Brain Heart Infusion (BHI) plates for quantifying bacterial loads.

## Mouse infections

Overnight cultures of the InlA* mutant LO28 strain of *L. monocytogenes* [18] were diluted to OD$_{600nm}$ 0.1 and cultured until OD$_{600nm}$ reached a value of 1. Then the bacterial pellets were

washed and diluted in PBS. $10^6$ bacteria calculated as above were used to intraperitoneally infect the 8-10-week-old mice. For survival assays, the weights of mice were monitored for 10 days after infection and the mice that lost 20% of their weight were sacrificed due to arrival at the humane endpoint. For CFU assays, spleens and livers were harvested three days after infection, homogenized in PBS and four 1:10 serial dilutions of these homogenates were plated on Oxford agar (Merck, #107004) supplemented with Listeria-selective supplement (Merck, #107006) for quantifying bacterial loads. For the FAO inhibition experiment, 20mg/kg etomoxir or corresponding volume of PBS was injected intraperitoneally into mice one day prior to and post-infection. Etomoxir-treated and PBS-treated mice were held in separate cages. Large lobe of the liver and spleen of the mice were harvested three days post-infection. For measurement of blood glucose, ALT and cytokines, the blood was withdrawn via heart puncture in EDTA tubes and spun at 13000 g for 5 min to obtain the plasma. ALT was measured by Labor Invitro GmbH (Vienna, Austria). Plasma cytokines were measured using Luminex-based, customized mouse 15-plex Invitrogen Procartaplex Immunoassay for mix&match panels according to the manufacturer's instructions.

## RNA sequencing and analysis

30 mg liver tissue from PBS-treated or one-day infected mice were shock-frozen in liquid nitrogen and crushed using a mortar and pestle. Tissues were further homogenized with a syringe and needle and transferred into ice-cold tubes for RNA isolation using Allprep DNA/RNA micro kit (Qiagen, #80284) according to manufacturer's instructions. The experiment was performed using three biological replicates per condition. The quality controls, RNA sequencing and preliminary analysis were performed by the Biomedical Sequencing Facility (CeMM, Vienna, Austria). Reads were mapped with TopHat to mm10 genome (RefSeq UCSC Mouse Dec. 2011/GRCm38/mm10 assembly). Cufflinks tool was used to assemble transcripts, estimate abundances and for differential expression analysis. All the RNA sequencing data is available in NCBI GEO repository on series record GSE 162448. GSEA analysis was performed with GSEA 4.0.3 using these FPKM values. Dot plots were generated with R software (version 4.0.2) using signal-to-noise (s2n) ratio for every gene set. The s2n was calculated by substracting the means of the gene counts per condition. This result was then divided by the sum of the deviations per condition. P value adjustment was done using the Benjamini hochberg procedure and a p value cutoff of 1 was used. Venn diagrams were generated using Venny 2.1 with the gene lists obtained from the GSEA analysis.

## Metabolite measurements

150 mg liver tissue was shock-frozen in liquid nitrogen and homogenized in 1.5 ml ice-cold HPLC grade MeOH:ACN:$H_2O$ (2:2:1, v/v) with tissue homogenizer at 30000 rpm for 15 sec. The homogenates were incubated on ice for 5 min and lysed further with two freeze-thaw cycles (5 min liquid nitrogen, 3 min 37°C). The lysates were centrifuged at 4000 g for 10 min at 4°C and supernatants were used for metabolite measurement by the Vienna Biocenter Metabolomics Core Facility. LC-MS data were subjected to statistical analysis using R software. The data are corrected for mouse sex, liver weight and experimental variation and ANOVA was used for analysis.

For macrophage metabolites $10^7$ BMDMs were scraped and washed in PBS and cell pellets were vortexed in 1 ml ice-cold HPLC grade MeOH:ACN:$H_2O$ (2:2:1, v/v) for 30 sec. Samples were exposed to two vortex-freeze-thaw-sonication cycles (1 min liquid nitrogen, 10 min thaw at RT sonicator) and incubated at -20°C for 1 hour for precipitating proteins. Following centrifugation at 13000 rpm at 4°C for 15 min, the supernatants were used for targeted metabolite

measurement performed by the Vienna Biocenter Metabolomics Core Facility. The precipitated proteins were resuspended in loading buffer (6M urea in 1.5x Laemmli buffer), 20 μl of this resuspension was run in 10% SDS-PAGE for western blotting. Tubulin was detected using anti-tubulin antibody diluted 1:5000 in 2% BSA (Sigma, #T9026) incubated 2 hours at RT followed by incubation in mouse secondary antibody (1:5000 in TBST) for 1 hour at RT. Tubulin band intensities, quantified using Image J64 (version 1.48), were used for normalization of metabolites.

## RNA isolation, cDNA synthesis and qPCR

Total RNA was isolated using NucleoSpin RNA kit (Macherey-Nagel, #740955), according to manufacturer's instructions. For liver tissue, one lobe of liver was homogenized with tissue homogenizer in 700 ml RA1 buffer whereas for BMDMs $10^6$ BMDMs were lysed in 350 ml buffer. The cDNA was generated using 400 ng RNA with oligo (dT18) primer and the RevertAid Reverse Transcriptase (Thermo-Fisher Scientific). Quantitative real-time PCR was performed using GoTaq master mix (Promega, #A600A). qPCR reactions were run on Mastercycler (Eppendorf). B2m and actin genes were used as housekeeping gene controls for liver and BMDMs, respectively. qPCR primers used in this study are listed in **S7 Table**.

## Seahorse flux analysis

A Seahorse XFe96 Analyzer (Agilent) was used to determine oxygen consumption rate (OCR) and extracellular acidification rate (ECAR) measurements of cultured cells in real-time. The Seahorse XF Cell Mito Stress test kit (Agilent, #103015–100) was used according to the manufacturer's instructions. BMDMs were infected with *L. monocytogenes* MOI 10 for 6 hours. After measurement of basal OCR and ECAR, oligomycin (1.5 μM), carbonyl cyanide-*p*-trifluoromethoxyphenylhydrazone (FCCP, 1.5 μM) and rotenone/antimycin A (0.5 μM) were injected subsequently. 8 μM Hoechst 33342 (Invitrogen, #H3570) was injected simultaneously with rotenone/antimycin A from a separate port. For FAO and glutaminolysis inhibition experiments, 4 μM Etomoxir and 3 μM BPTES (Agilent, #103260–100) were injected after FCCP instead of Hoechst. The raw data was analysed using Wave Desktop Software (Agilent, version 2.6.1) and exported and graphed in GraphPad Prism 7 (GraphPad Software).

## Staining for flow cytometry analysis

For cell recruitment assays in infected mice, peritoneal exudates were collected by injecting 5 ml PBS into the belly of sacrificed mice twice. After counting, the cells were stained for dead cells using fixable viability dye (eBioscience, #65-0865-14). Then, cells were washed in FC buffer (0.5% BSA in PBS) and blocked using $F_c$g block at RT for 10 min and stained in FC buffer (0.5% BSA in PBS) at RT for 30 min. Cells were washed with PBS and analyzed by flow cytometry using using BD LSRFortessa. For splenocyte staining, the spleens harvested from mice were passed through the 70 mm cell strainer in 5 ml PBS and exposed to red blood cell lysis buffer (150 mM $NH_4Cl$, 10 mM $KHCO_3$, 0.1 mM $Na_2EDTA$, pH 7.3) for 5 min on ice. Then cells were washed with PBS twice and stained with the viability dye and antibodies as above. Antibodies used for flow cytometry are CD3e (BD, #562286), B220 (BD, #553090), Ly6G (BD, #561236), CD11b (BD, #563015), Ly6C (BD, #560594). Flow cytometry was performed using BD LSRFortessa. FlowJo 10.6.1 software was used to analyze the data.

## Western blot and its quantification

Infected and gentamicin treated BMDMs were washed with PBS and lysed in Laemmli buffer (2% SDS, 10% glycerol, 62.5 mM Tris, pH 6.8). Protein concentration was determined using

BCA assay (ThermoFisher, #23225) and after adding 5% β-mercaptoethanol and bromophenol blue (BPB), the samples were loaded in 10% SDS-PAGE gel.

32.5 mg liver tissue from infected mice was homogenized per ml lysis buffer (10 mM Tris pH7.5, 50 mM NaCl, 50 mM NaF, 2 mM EDTA, 1% Triton X-100, 1 mM DTT, 0.1 mM PMSF, 1 mM Vanadate, cOmplete protease inhibitor cocktail tablet (Roche, 11697498001), 10 mg/ml DNase I). Then the lysates were passed through a 25G syringe (Braun, #9186166) three times, incubated at 4˚C on a rotating wheel for 2 hours and spun at 13400 g at 4˚C for 20 min. The supernatant was transferred into a new tube and kept at -20˚C. Lysates were then mixed with Laemmli buffer for loading on a 10% SDS-PAGE gel.

The samples were loaded twice to detect the phospho- and total forms of the proteins separately. The gels were blotted on nitrocellulose membrane for 16 hours at 200 mA and then 2 hours at 400 mA at 4˚C in transfer buffer (3 mM $Na_2CO_3$, 10 mM $NaHCO_3$, and 20% ethanol). The membranes were blocked in 5% non-fat dry milk powder in TBST for 1 hour at RT and incubated with primary antibodies detecting total ACC (CST, #3662) or phospho-ACC (CST, #3661) prepared in 5% BSA in 1:1000 dilution for overnight at 4˚C while shaking. Next day membranes were washed in TBST three times for 5 min and incubated at RT with rabbit secondary antibody (Jackson Immunoresearch, #111-035-003) in 1:2500 dilution for BMDMs and 1:10000 dilution for liver lysates. For development of ACC and phospho-ACC signals SuperSignal West Pico PLUS (Thermo Scientific, #34580) and ECL Prime Western Blotting Detection Reagent (Amersham, #RPN2236) were used. For detection, the BioRad ChemiDoc imaging system was used.

Quantification of western blots were done with bands in the linear detection range. Serial lysate dilutions used were for BMDMs 1.25 mg, 2.5 mg, 5 mg, 10 mg, 20 mg and for liver tissue lysates 0.3125 ml, 0.625 ml, 1.25 ml, 2.5 ml and 5 ml. Minimum of three of these dilutions were used for the downstream calculations. Slope ratios of phospho-ACC and total ACC were calculated for all genotypes and the knockouts were normalized to wt.

## Statistics

All statistical analysis was performed in GraphPad Prism 7. Statistical tests used for the calculation of p values are indicated in the figure legends.

## Statistical analysis of liver metabolites

Statistical analysis was performed in R version 3.6.2 [40]. We fitted univariate linear models using function *L. monocytogenes* for each log10 transformed measured metabolite in turn. Effects of genotype and infection and the interaction between them are of main biological interest in our models. Both were modelled as fixed categorical effects with three levels each; for genotype (wt, *Irf9*$^{-/-}$ and *Ifnar1*$^{-/-}$) and infection (d0, d1, d3), respectively. Dummy coded and centered sex of the mice, dummy coded and centered batch/experiment and z-transformed liver weights were fitted as cofactors to reduce residual variability. Assumptions for linear models were met. Residuals were normally distributed and homoscedastic. We calculated contrasts between least square means (LSM) of genotype and infection levels respectively with package *emmeans* v1.4.7 [41]. Significance was declared at a multiple testing corrected 10% false discover rate. [42].

We further performed Principal Component Analysis with package *factoextra* v1.0.7 [43]. We produced biplots using function *fviz_pca_biplot*, which display PCA scores of samples (shown as dots) and loadings of each metabolite (shown as vectors) simultaneously. Dots that are close to each other represent mice with similar values. The longer a vector of a metabolite the bigger the influence of said metabolite on that principal component. Vectors pointing in

similar directions, forming small angles between them can be considered as positively corre-
lated, vectors forming an angle of 90˚ as uncorrelated and vectors pointing in opposing direc-
tions as negatively correlated. We used centered and scaled log10 transformed metabolite
measures corrected for batch / experiment effects, liver weight and sex for our PCA. We calcu-
lated these 'customized' residuals by subtracting estimates for each experiment, sex and liver
weight from the same linear model as used for hypothesis testing, from the raw metabolite
measures. Numbers of mice used for this screen are for uninfected wt, *Irf9*$^{-/-}$ and *Ifnar1*$^{-/-}$, 15,
15 and 13; for one-day infected wt, *Irf9*$^{-/-}$ and *Ifnar1*$^{-/-}$, 15, 15 and 15; for three-day infected wt,
*Irf9*$^{-/-}$ and *Ifnar1*$^{-/-}$, 13, 14 and 14.

## Supporting information

**S1 Fig. (Related to Fig 1).** (A) Cytokine levels of mice uninfected or infected for one-day. (B)
Neutrophil recruitment into the peritoneal cavity of PBS-treated mice or one day post-infec-
tion. (C) Gating strategy for flow cytometry of PECs in Figs 1 and S1. P values were calculated
with two-way ANOVA test.
(TIF)

**S2 Fig. (Related to Fig 2).** (A) GSEA enrichment plots for OXPHOS and FA metabolism of
IFN-I treated wt and *Irf9*$^{-/-}$ BMDMs. (B) Extracted ion counts (EIC) of intracellular free FAs
normalized to tubulin (n = 3 per condition), (C) mRNA expression of *Cd36* (n = 3 per condi-
tion), (E) OCR and (F) ECAR in BMDMs 6h post-infection or medium treatment. (D) Three
replicates of Western blots used to quantify ACC phosphorylation in BMDMs 6h post-infec-
tion.
(TIF)

**S3 Fig. (Related to Fig 2).** Mean OCR of four replicates of (A) Etomoxir- and (B) BPTES-
treated uninfected BMDMs that are shown in Fig 2. The inhibitors were added at the time
point indicated by the arrow. OCR difference of (C) Etomoxir-treated and (D) BPTES-treated
BMDMs calculated by subtracting OCR at 73 min (addition of Rotenone/Antimycin) from
OCR at 53 min. OCR differences were calculated using OCR values measured at time points
indicated with dashed lines. Bars show the mean values. P values were calculated using
ANOVA corrected for multiple testing with Dunnett's post-hoc test.
(TIF)

**S4 Fig. (Related to Fig 3).** (A) Dot plots of GSEA results showing first 20 enriched pathways
in uninfected (A) *Irf9*$^{-/-}$ and *Ifnar1*$^{-/-}$ compared to uninfected wt. Color and size of the dots
represent adjusted p values and the number of the genes represented from a certain gene set,
respectively. The x-axes were calculated by the sum of the core-enriched genes divided by its
set size and ordered decreasingly. The y-axis represents the corresponding gene set name. (B)
Venn diagram showing the number of genes enriched in uninfected *Irf9*$^{-/-}$ and *Ifnar1*$^{-/-}$ com-
pared to uninfected wt.
(TIF)

**S5 Fig. (Related to Fig 3).** Quantitative RT-PCR of (A) OXPHOS, (B) FA metabolism and (C)
IFN-stimulated genes from uninfected and one-day infected mouse livers. P values were calcu-
lated using ANOVA corrected for multiple testing with Dunnett's post-hoc test.
(TIF)

**S6 Fig. (Related to Fig 5).** (A-B) Loading plots showing the contribution of each metabolite to
clustering with vectors. Length and color of the vectors represent their contribution to separa-
tion of different clusters. Impact of each metabolite on a certain cluster separation is reflected

in the direction of the vectors. Percentage of total variance is indicated in the axis label. The values are corrected for liver weight, genotype, sex of the mouse and time point effects. ANCOVA test was performed. Extracted ion counts (EIC) of intracellular (C) FAs and (D) ketone bodies in wt, *Irf9*[-/-] and *Ifnar1*[-/-] mice one- or three-days post-infection or PBS injection. False discovery rate p values were calculated using values corrected for liver weight, genotype, sex of the mouse and time point effects. ANCOVA test was performed. (E) Scheme showing glutaminolysis replenishing the TCA cycle. (F) mRNA expression of glutamate carrier *Slc25a22* in uninfected wt, *Irf9*[-/-] and *Ifnar1*[-/-] livers. P values were calculated using ANOVA with Tukey post-hoc test. (G) Three replicates of Western blots used to quantify ACC phosphorylation in livers one day post-infection. Contrib.: contribution, PC: Principal component, PGA: Phosphoglyceric acid, PEP: Phosphoenolpyruvate, Asp: Aspartic acid, AcCoA: Acetyl CoA, Glu: Glutamate, aKG: alpha-ketoglutarate, BHOB: Hydroxybutyric acid, SA: Stearic acid, AcAct: Acetoacetate, Val: Valine, Leu: Leucine, Ile: Isoleucine, PA: Palmitic acid, OA: oleic acid, Ser: Serine, POA: Palmitoleic acid, aGlc: alpha-D-glucose, bGlc: beta-D-glucose, Pyr: Pyruvate, G6P: Glucose-6-phosphate, GA3P: Glyceraldehyde-3-phosphate.
(TIF)

**S1 Table. GSEA of RNAseq from livers showing enrichment in uninfected *Irf9*[-/-] compared to uninfected wt.**
(XLSX)

**S2 Table. GSEA of RNAseq from livers showing enrichment in uninfected *Ifnar1*[-/-] compared to uninfected wt.**
(XLSX)

**S3 Table. GSEA of RNAseq from livers showing enrichment in infected wt compared to uninfected wt.**
(XLSX)

**S4 Table. GSEA of RNAseq from livers showing enrichment in infected *Irf9*[-/-] compared to infected wt.**
(XLSX)

**S5 Table. GSEA of RNAseq from livers showing enrichment in infected *Ifnar1*[-/-] compared to infected wt.**
(XLSX)

**S6 Table. List of genes in the Venn diagrams in Fig 3E.**
(XLSX)

**S7 Table. List of primers used in this study.**
(XLSX)

## Acknowledgments

Metabolomics analysis was performed by the Metabolomics Facility at Vienna BioCenter Core Facilities (VBCF), member of the Vienna BioCenter (VBC), Austria and funded by the City of Vienna through the Vienna Business Agency. We would like to especially thank Thomas Kocher for data discussions and suggestions. We also would like to thank to Pavel Kovarik and Levent Bas for critical reading of our manuscripts and sharing their feedback.

## Author Contributions

**Conceptualization:** Duygu Demiroz, Thomas Decker.

**Data curation:** Duygu Demiroz, Philipp Fischer, Marlies Dolezal.

**Formal analysis:** Duygu Demiroz, Marlies Dolezal.

**Funding acquisition:** Duygu Demiroz, Manuela Baccarini, Mathias Müller, Veronika Sexl, Thomas Decker.

**Investigation:** Duygu Demiroz, Ekaterini Platanitis, Michael Bryant, Michaela Prchal-Murphy, Alexander Lercher, Caroline Lassnig.

**Methodology:** Duygu Demiroz, Michaela Prchal-Murphy.

**Project administration:** Duygu Demiroz, Thomas Decker.

**Resources:** Caroline Lassnig, Mathias Müller, Andreas Bergthaler, Veronika Sexl, Thomas Decker.

**Software:** Marlies Dolezal.

**Supervision:** Manuela Baccarini, Mathias Müller, Andreas Bergthaler, Veronika Sexl, Thomas Decker.

**Validation:** Duygu Demiroz, Marlies Dolezal.

**Visualization:** Duygu Demiroz, Philipp Fischer.

**Writing – original draft:** Duygu Demiroz, Thomas Decker.

**Writing – review & editing:** Ekaterini Platanitis, Andreas Bergthaler, Veronika Sexl.

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
