## [Decision Letter · Decision Letter 0]

28 Jan 2021

Dear Prof. Decker,

Thank you very much for submitting your manuscript "Listeria monocytogenes infection rewires host metabolism with regulatory input from type I interferons" for consideration at PLOS Pathogens. As with all papers reviewed by the journal, your manuscript was reviewed by members of the editorial board and by several independent reviewers. In light of the reviews (below this email), we would like to invite the resubmission of a significantly-revised version that takes into account the reviewers' comments.

Overall, the reviewers concurred that your study focused on an interesting and important question, and were generally positive about the data supporting a role for IRF9 and IFNAR in modulating host metabolism during infection. While a number of specific suggestions were provided by the reviewers that would improve the manuscript, two major points should be substantively addressed in any future resubmission.

First, two reviewers noted that while your data supported the conclusions that IRF9 and IFNAR regulated susceptibility to infection, and also contributed to re-wiring of host metabolism, the relationship between metabolic changes and susceptibility to infection was unclear. The study would greatly benefit from experimental approaches that could test the role of one or more of the altered metabolites in susceptibility of these mice to Listeria infection. Second, validation of prominent hits from the transcriptomics data would enhance the rigor of the study.

If you choose to resubmit a revised manuscript, please also be attentive to comments from Reviewer #1 regarding concerns about the effects of high gentamicin on viability of intracellular bacteria, and from Reviewer #2 regarding data visualization and readability.

We cannot make any decision about publication until we have seen the revised manuscript and your response to the reviewers' comments. Your revised manuscript is also likely to be sent to reviewers for further evaluation.

Sincerely,

Mary O'Riordan

Associate Editor

PLOS Pathogens

Raphael Valdivia

Section Editor

PLOS Pathogens

Kasturi Haldar

Editor-in-Chief

PLOS Pathogens

orcid.org/0000-0001-5065-158X

Michael Malim

Editor-in-Chief

PLOS Pathogens

orcid.org/0000-0002-7699-2064

Overall, the reviewers concurred that your study focused on an interesting and important question, and were generally positive about the data supporting a role for IRF9 and IFNAR in modulating host metabolism during infection.

While a number of specific suggestions were provided by the reviewers that would improve the manuscript, two major points should be addressed with additional evidence in any future resubmission.

First, two reviewers noted that while your data supported the conclusions that IRF9 and IFNAR regulated host metabolism and also regulated susceptibility to infection, the causal relationship between metabolic changes and susceptibility to infection was not established. The study would greatly benefit from experiments supporting a role for one or more of the identified metabolites in the susceptibility of these mice to Listeria infection. Second, validation of prominent hits from the transcriptomics data would enhance the rigor of the study.

If you choose to resubmit a revised manuscript, please also be attentive to comments from Reviewer #1 regarding concerns about the effects of high gentamicin on viability of intracellular bacteria, and from Reviewer #2 regarding data visualization and readability.

Reviewer's Responses to Questions

**Part I - Summary**

Reviewer #1: Interferon response to infection and immunometabolism are very important areas of research. In this manuscript Demiroz et al. investigate the role of IFN-I in metabolic processes of murine macrophages and liver upon Listeria monocytogenes infection. Using genetically-modified mice and transcriptomic and metabolomic profiling, they uncover a very interesting specificity of IRF9 and IFNAR in regulating host metabolism and susceptibility to infection.

Reviewer #2: In the enclosed manuscript from Demiroz et al the authors investigate the role of type I interferons in modulating metabolism in the context of Listeria monocytogenes infection. Initially they recapitulate much of what has been reported in the literature with respect to loss of interferon signaling, including decreases in liver damage and decreased lymphocyte apoptosis in IFN signaling deficient animals despite minor effects on bacterial burden. Importantly, one novel observation from this work is that IRF9 mutants are even more protected than IFNAR mutants. The authors then go on to use bone marrow derived macrophages to assess the impact of IFNAR signaling on metabolic pathways. They find that ACC phosphorylation is decreased and less FA is imported into the mitochondria in the absence of IFNAR or IRF9. Consistent with this they observe decreased OXPHOS as measured by seahorse, although the contribution of IRF9 was significantly less than that of IFNAR suggesting IRF9 independent pathways downstream of IFNAR. This is confusing as the protection phenotypes presented in Fig. 1 were more prominent in the IRF9-/- than IFNAR-/-. Next the authors attempted to assess whether similar changes take place in vivo during infection and utilized transcriptomics to assess hepatic responses to Listeria in wt, IRF9 and IFNAR deficient mice. Unfortunately the data presented in figure 3 is incredibly confusing as it appears OXPHOS and FA metabolism are even further upregulated in IFNAR and IRF9 which is inconsistent with the metabolism described in figure 2. The authors do make the point that the differential regulated transcripts of IFNAR and IRF9 do not completely overlap again suggesting differential signaling downstream of IFNAR, however it is unclear to me what the data in figure 3 tells us as currently presented. As the data from figure 3 was inconclusive, the authors directly assessed a subset of metabolites from infected livers from the three mice genotypes. Again the data presentation in figure 4 is difficult to assess and as the authors themselves describe them “these data show that L. monocytogenes infection and IFN-I signaling cause metabolic changes in the liver”, a conclusion that does not get at the differences in susceptibility to infection or even really the role of Listeria infection or type I interferons in regulating metabolism. Finally, the data in figure 5 are largely in agreement with the macrophage data from figure 2 demonstrating decreases in ACC phosphorylation and FA transport into the mitochondria, although again these changes do not correlate with the differences in susceptibility reported in figure 1, notably increased resistance of IRF9-/- mice vs IFNAR-/-. In sum, while this manuscript aims to determine how IFN-I regulates metabolism in the context of Listeria infection to discern if this contributes to the differential susceptibility of mice with IFN signaling deficiencies, there is no conclusive data presented either about how IFN alters metabolism or whether or not this contributes to infection. Furthermore, while the observation that IRF9 mice are more protected from infection than IFNAR mice is interesting, there are no data presented to explain this phenomenon.

Reviewer #3: Demiroz and colleagues have tackled an important and outstanding question in Listeria biology and their work implicates a role for IFN signaling in the control of metabolic responses in both infected macrophages and liver tissue from infected animals. They have used a combination of metabolomics, RNA-seq and real time readouts of oxidative phosphorylation using a seahorse analysis to reveal a difference in the metabolism of Irf9 and IFNAR KO animals. This is particularly important since the reason for reduced bacterial load in IFNAR animals remains elusive. I very much enjoyed the paper and have several suggestions to revise it primarily for improved transparency and clarity.

**Part II – Major Issues: Key Experiments Required for Acceptance**

Reviewer #1: 1-Infection route. The puzzling choice of IP injection is all the more surprising since the authors have previously demonstrated that IFN-I have opposite roles in mice infected by the IP route or by the natural, more relevant oral route. Are the different contributions to infection control of IRF9 and IFNAR also seen in mice orally infected with the LO28InlA* strain?

2-Significance of metabolic effects. WT, Ifr9-/- and Ifnar1-/- mice show rather small metabolic differences. There is no evidence that these differences contribute to infection control. The study would greatly benefit from an in vivo experiment showing the role of some of the identified metabolites in the susceptibility of these mice to infection.

3-Gentamicin protection assays. Use of high concentrations of gentamicin such as 50 ug/mL for 1 hour leads to gentamicin internalization and bactericidal effects demonstrated in several infection models. What is the intracellular survival rate of the LO28 strain after 1 hour of infection and treatment with gentamicin 50 ug/mL compared to treatment with gentamicin 20, 10 or 5 ug/mL? Once the optimal concentration of gentamicin is identified, CFU/well (instead of fluorescence) should be determined at 2, 6 and 24 hours (if cell viability is not affected). A time "zero" control should be added to demonstrate that the capacity of internalization is similar in WT, Ifr9-/- and Ifnar1-/- cells.

Reviewer #2: Major Points:

1) The observation that IRF9 mice are even more resistant than IFNAR mice is interesting and novel but there is no work presented to explain this phenomenon. As the transcriptomics data in figure 3 focus only on potential metabolic differences which appear to not explain the phenotype, significant additional studies are necessary to dissect the reason for this phenotype.

2) Although not a major contributor to the story the authors try to tell, the cell intrinsic differences in macrophage control of Listeria infection presented in figure 1H are inconsistent with the existing literature that has found no cell intrinsic difference in IFNAR macrophage control of Listeria infection (ex. Auerbuch et al 2004). Why the authors see something different is not explained and this is important as these same cells are then used for all the metabolic analysis in figure 2 and it is critical to know if there is something different than other papers in the field.

3) The transcriptomics data are presented in a very confusing way and no validation of any of the data were performed. As presented the data is not useful.

4) Figure 4 suffers the same readability issues that as presented it is difficult to glean meaningful information from the data, as the authors themselves seem to concede in stating simply that there are differences in metabolites. This data is at least followed up with analysis of specific metabolites in figure 5.

5) Most importantly, there is no connection with any of the metabolic data presented in the paper and a causal relationship with the differential susceptibility of IFN signaling deficient mice. In the end the manuscript ends up as a list of data with no appreciation for the biological impact of any of it. The authors themselves state “the rather small metabolic effects observed in our macrophage experiments alone are unlikely to make major contributions to the substantial differences between Irf9-/- and Ifnar1-/- mice in the resistance to L. monocytogenes infection”. In the end, this is the crux of the issue, data was presented that there are metabolic differences in the knockout mice but the phenotypes are small and there is no indication that they are biologically relevant without additional experimentation to manipulate the pathways and assess subsequent Listeria infection outcomes.

Reviewer #3: Overall, the use of varied technology and analyses that go from primary cells in culture to in vivo experiments is impressive. Given the varied metabolic contribution of macrophages/circulating monocytes compared with hepatocytes do the authors hypothesize that the same pathways are at play in both conditions? The final conclusions seem to indicate that they suspect the same pathway but it seems like the RNA-seq data would have incorporated a mix of macrophages and hepatocytes whereas the BMDM data would represent metabolic data solely derived from immune cells. When the authors mined their previous BMDM data from Platanitis E, et al., how much does it overlap with the RNA-seq from this analysis? Do they find the same proteins up or down?

Along these lines could they infect animals and separate out immune cells from hepatocytes and analyze levels of p-ACC or GABA to access whether the immune cells and hepatocytes have the same response? They could also add infection of primary hepatocytes or a hepatocyte cell line to sort this out. I’m not sure if there is a way to assess GABA in a less costly experiment than metabolomics, but the overarching question is how general is the response that they describe (same in all infected cells) or could it be different in the macrophages versus hepatocytes? If the authors expect both immune cells and hepatocytes to act in the same manner, this should just be more fully discussed throughout the text and highlighted on the figures (where the source material comes from, while the figure legends and text indicate the source material, it would be better to see it on the figures as well). Adoptive transfer experiments or tissue specific deletion (hepatocyte versus immune CRE) could address this issue elegantly but could be reserved for future work.

The authors spend a fair amount of time in the introduction explaining the role of various metabolic pathways in activated macrophages. Are BMDMs the best model to study this phenomenon or would resident or peritoneal macrophages be a better representation of the monocytes that they are interested in in vivo? How would polarization or pretreatment to activate BMDMs affect or change the phenotype?

The authors should upload their RNA-seq data to a searchable database or upload the log transformed data in the form of an excel table that highlights the individual proteins detected and their values. They have done this with the enrichment analyses but unless I missed it the complete RNA-seq analysis is not part of the supplementary data.

The authors did not show the raw Western blot data for phospho ACC levels. At least a representative blot should be shown, but it would be preferable to see all three in supplementary data in line with increased transparency.

**Part III – Minor Issues: Editorial and Data Presentation Modifications**

Reviewer #1: 1-At day 3, injection of 10e6 LO28 led to death of some WT and Ifnar1-/- mice, and thus heterogeneity. It could be worth measuring bacterial survival upon infection with a sublethal inoculum.

2-Many phenotypes are investigated in WT, Ifr9-/- and Ifnar1-/- backgrounds with or without infection. A summary of what is IRF9-dependent, IFNAR-dependent, IRF9- and IFNAR-dependent and IRF9- and IFNAR-independent would be helpful in the discussion. For instance, figure 6 could be completed with an Ifr9-/- cell, liver data and uninfected data.

3-Line 159 It is not clear why the Wang reference is introduced here.

4-Line 396 The mice genotype should be indicated.

5-Fig S1C is not introduced.

6-Some references should be completed (journal, page numbers etc.).

7-Western blots should be shown (Figures 2 and 5)

8-It would help to indicate clearly on figures what is related to uninfected or infected data.

9-The unacceptable quality of the main figures, probably due to file conversion, should be improved. Figures 3 and 4 are partially illegible...

Reviewer #2: (No Response)

Reviewer #3: Discussion points/Figure revision

In their previous work the authors often compared IRF3/7 KO animals to IFNAR. It is beyond the scope of this paper to cross compare those cells/animals but I wonder if they would discuss what they would expect? Would cytosolic DNA/RNA signaling that is IRF9-independent potentially be responsible for this phenotype observed in the IFNAR KO? Along these lines, when they cross compare IFNAR and IRF9 KO mice they have overlapping genes in Venn diagrams but I was not able to find the list of these genes. Are they in a table that I missed?

For the PCA analysis, I think the figure should be revised so one can see the general distribution of the phenotypes and the individual differences in pathways that they wish to highlight could be displayed as bar graphs below. As it stands, it is very difficult to make out the various conditions due to the labeling.

The authors have included diagrams of the metabolic pathways that they discuss in the supplementary figures but it may be worthwhile to incorporate such diagrams into the figures so that readers can cross compare

PLOS authors have the option to publish the peer review history of their article (what does this mean?). If published, this will include your full peer review and any attached files.

Reviewer #1: No

Reviewer #2: No

Reviewer #3: No
---

## [Decision Letter · Decision Letter 1]

7 Jun 2021

Dear Prof. Decker,

We are pleased to inform you that your manuscript 'Listeria monocytogenes infection rewires host metabolism with regulatory input from type I interferons' has been provisionally accepted for publication in PLOS Pathogens.

Best regards,

Mary O'Riordan

Associate Editor

PLOS Pathogens

Raphael Valdivia

Section Editor

PLOS Pathogens

Kasturi Haldar

Editor-in-Chief

PLOS Pathogens

orcid.org/0000-0001-5065-158X

Michael Malim

Editor-in-Chief

PLOS Pathogens

orcid.org/0000-0002-7699-2064

Reviewer Comments (if any, and for reference):

Reviewer's Responses to Questions

**Part I - Summary**

Reviewer #2: In the revised manuscript by Demiroz et al the authors aimed to address concerns about the lack of mechanistic understanding behind their interesting observations that IRF9 deficient mice are protected from listeriosis, even more so that the protection previously described for IFNAR deficient mice. While previous rounds of review noted the significant amount of work presented in the manuscript, there were concerns that the data neither clearly indicated how IFN manipulated metabolism nor how this putative manipulated metabolism contributed to the resistance phenotype of the knockout mice. In the revised manuscript, the authors have added data including important validation of transcriptomic data by qPCR for example. In addition the investigators attempted to directly test the hypothesis that regulation of FAO is a mechanism by which IFN contributes to control of Listeria infection by treating mice with etomoxir, however the interpretation of this data is difficult. The investigators see a minor decrease in burden in WT mice but not in IRF9 or IFNAR mice with etomoxir treatment which they interpret as a consistent with their model that IFN regulation of FAO contributes to control of Listeria burden. This data is overinterpreted as all they have shown is that FAO inhibition contributes to infection control but there is no data to conclude that this is the mechanism observed in IFNAR or IRF9 knockout mice. Additionally, the investigators ignore the increase in burden observed in the IFNAR knockout mice that would not be consistent with this model? Furthermore, although they have tried to clarify some of the PCA analysis in the end the metabolomics are incongruent with the transcriptomics. The data are the data so this is fine, however the problem is that none of the newly added data does anything to clarify how IFN regulates metabolism to control infection or why IRF9 mutants would behave differently than IFNAR mutants. As noted by the authors themselves both the metabolic changes and many of the transcriptional changes are very small in magnitude and perhaps this is the reason for significant lack of overlap between the IRF9 and IFNAR data that is surprising and unexpected. I appreciated the authors discussion of why IRF9 might be different than IFNAR, however not following up on these ideas ultimately leads to a paper that is simply an interesting observation that remains unexplained.

Reviewer #3: This paper maps the role of IFNAR and IRF9 on metabolic reprogramming by Listeria monocytogenes and will be a valuable contribution to an interesting and up and coming field.

**Part II – Major Issues: Key Experiments Required for Acceptance**

Reviewer #2: (No Response)

Reviewer #3: I am satisfied with the revised manuscript. The reviewers addressed my concerns and I think the revision is ready for publication.

**Part III – Minor Issues: Editorial and Data Presentation Modifications**

Reviewer #2: (No Response)

Reviewer #3: The authors use IFN-I as a plural noun throughout, perhaps it would be better to specifically say Type I interferon. I thought it is more grammatically appropriate to use Type I IFN in the singular; e.g.

Line 102 “IFN-I are integral” – Type I IFN is integral.

I think the acronym is making the grammar complicated.

PLOS authors have the option to publish the peer review history of their article (what does this mean?). If published, this will include your full peer review and any attached files.

Reviewer #2: No

Reviewer #3: No

---

## [Editor Report · Acceptance letter]

24 Jun 2021

Dear Prof. Decker,

We are delighted to inform you that your manuscript, "Listeria monocytogenes infection rewires host metabolism with regulatory input from type I interferons," has been formally accepted for publication in PLOS Pathogens.

Best regards,

Kasturi Haldar

Editor-in-Chief

PLOS Pathogens

orcid.org/0000-0001-5065-158X

Michael Malim

Editor-in-Chief

PLOS Pathogens

orcid.org/0000-0002-7699-2064